# Exploitation of a Latent Mechanism in Graph Contrastive Learning: Representation Scattering

**Dongxiao He[1], Lianze Shan[1], Jitao Zhao[1], Hengrui Zhang[2], Zhen Wang[3]*, Weixiong Zhang[4]**

[1]College of Intelligence and Computing, Tianjin University, Tianjin, China
[2]Department of Computer Science, University of Illinois at Chicago, Chicago, IL, United States
[3]School of Cybersecurity, Northwestern Polytechnical University, Xi'an, China
[4]Department of Computing, Department of Health Technology and Informatics,
The Hong Kong Polytechnic University, Kowloon, Hong Kong
[1]{hedongxiao, shanlz2119, zjtao}@tju.edu.cn,
[2]hzhan55@uic.edu, [3]w-zhen@nwpu.edu.cn, [4]weixiong.zhang@polyu.edu.hk

## Abstract

Graph Contrastive Learning (GCL) has emerged as a powerful approach for generating graph representations without the need for manual annotation. Most advanced GCL methods fall into three main frameworks: node discrimination, group discrimination, and bootstrapping schemes, all of which achieve comparable performance. However, the underlying mechanisms and factors that contribute to their effectiveness are not yet fully understood. In this paper, we revisit these frameworks and reveal a common mechanism—representation scattering—that significantly enhances their performance. Our discovery highlights an essential feature of GCL and unifies these seemingly disparate methods under the concept of representation scattering. To leverage this insight, we introduce Scattering Graph Representation Learning (SGRL), a novel framework that incorporates a new representation scattering mechanism designed to enhance representation diversity through a center-away strategy. Additionally, consider the interconnected nature of graphs, we develop a topology-based constraint mechanism that integrates graph structural properties with representation scattering to prevent excessive scattering. We extensively evaluate SGRL across various downstream tasks on benchmark datasets, demonstrating its efficacy and superiority over existing GCL methods. Our findings underscore the significance of representation scattering in GCL and provide a structured framework for harnessing this mechanism to advance graph representation learning. The code of SGRL is at https://github.com/hedongxiao-tju/SGRL.

## 1 Introduction

Graph Neural Networks (GNNs) have shown impressive performance across various fields, including social networks [1, 2], bioinformatics [3, 4], and fraud detection [5]. However, training GNNs typically requires large datasets with manually annotated labels, which can be both costly and labor-intensive [6]. This limitation restricts their broader application. To address this challenge, Graph Contrastive Learning (GCL) has attracted significant attention [7, 8, 9, 10], focusing on creating proxy tasks from the data itself to enable self-supervised training of GNN encoders [11]. Currently, most GCL research [6, 12, 13] is concentrated on enhancing one of the three main frameworks: InfoNCE-based (i.e., node discrimination) [14], DGI-like (i.e., group discrimination) [7], and BGRL-like (i.e., bootstrapping schemes) [10].

---

*Corresponding author

38th Conference on Neural Information Processing Systems (NeurIPS 2024).

The three mainstream graph contrastive learning frameworks differ significantly, particularly in their approach to node-level tasks. InfoNCE-based methods prioritize node-level discrimination [8, 9, 12], treating an anchor node as a reference while considering all other nodes as negative samples to enhance the distinctiveness of each node's representation. In contrast, DGI-like methods adopt a group discrimination paradigm [6, 7, 15], viewing all nodes as positive samples from the same distribution and differentiating them from a noise distribution. BGRL-like methods [10, 13, 16] employ a bootstrapping scheme that eliminates the need for negative samples during training, focusing instead on aligning positive samples. Despite these differences, all three frameworks demonstrate comparable performance, leading us to conjecture **that they may share a common mechanism.** This idea is partially inspired by previous research in visual contrastive learning, which identified uniformity as a key factor in many contrastive learning methods [17]. However, uniformity, which focuses solely on the discriminative properties of node instances, cannot explain the shared underlying factors across these GCL frameworks, as further discussed in Appendix A.

We carried out a detailed analysis of the three GCL frameworks and discovered that representation scattering is a crucial common factor in their success (in Section 3). In Section 3, we provide a formal definition of representation scattering and examine how each framework achieves it. For the DGI framework, we analyze the distributions of original and noise data, proving that after GNN message passing [18, 19], the noise data distribution aligns with the mean distribution of the original data. This indicates that DGI's objective can be interpreted as distinguishing between the local semantics of nodes within the original graph and its mean, which correlates with representation scattering (Section 3.1). In the InfoNCE framework, existing work has demonstrated that the mechanism of negative sampling facilitates a process of achieving uniformity [17]. We extend this further by theoretically proving that the InfoNCE loss serves as an upper bound for representation scattering loss (Section 3.2). Regarding the BGRL framework, we explore the role of Batch Normalization [20] and its connection to representation scattering, showing that it acts as a specific instance of this concept. Notably, removing Batch Normalization significantly degrades BGRL's performance (Figure 2). Our findings confirm that representation scattering is a key mechanism present across all three mainstream GCL frameworks.

However, the existing GCL frameworks have not fully leveraged the latent mechanisms inherent in their designs and neglected the interconnected nature of graphs when implementing representation scattering. This oversight results in inefficiencies and reduced robustness. Firstly, generating augmented views or computing similarities between node pairs for representation scattering [7, 8, 9] incurs significant computational and memory overhead. Secondly, manually defined negative samples may result in the generation of numerous false negative samples, introducing noise [12] that can hinder model training. Lastly, those frameworks that rely solely on positive samples [10] implement representation scattering indirectly through Batch Normalization, which lacks explicit guidance for scattering and can lead to suboptimal performance. Addressing these challenges is crucial for enhancing the effectiveness of GCL methods.

To fully and effectively utilize representation scattering, we propose **S**cattering **G**raph **R**epresentation **L**earning (SGRL). Our approach introduces a Representation Scattering Mechanism (RSM) that embeds node representations into a designated hypersphere, positioning them away from the mean center. This method offers a direct algorithm for representation scattering compared to the existing GCL frameworks, eliminating biases introduced by manually defined negative samples. Additionally, we introduce a Topology-based Constraint Mechanism (TCM) that considers the interconnected nature of graphs. TCM aligns representations derived from structural information with scattered representations, thereby preserving topology information while facilitating scattering. Through these innovations, we aim to enhance the efficiency and robustness of GCL methods. We made the following contributions in this paper:

- We discovered a common representation scattering mechanism in GCLs and showed that the three mainstream GCL frameworks implicitly utilize this mechanism, and importantly can be unified under the concept to representation scattering.

- We showed that the existing methods do not fully exploit the inherent mechanism of representation scattering. We introduced the novel SGRL framework to integrate RSM and TCM, producing an adaptive scattering approach for model training.

- We experimented with various downstream tasks on benchmark datasets and demonstrated the effectiveness and efficiency of SGRL.

## 2 Preliminary

**Graph Data.** We define a graph as $\mathcal{G} = (\mathcal{V}, \mathcal{E})$, where $\mathcal{V} = \{v_1, v_2, \ldots, v_N\}$ denotes the set of nodes, and $\mathcal{E} \subseteq \mathcal{V} \times \mathcal{V}$ represents the set of edges. The node feature matrix is denoted by $\mathbf{X} \in \mathbb{R}^{N \times D}$, where $N$ is the number of nodes and $D$ is the feature dimension. In addition, the adjacency matrix is indicated by $\mathbf{A} \in \mathbb{R}^{N \times N}$, formulated such that $\mathbf{A}_{ij} = 1$ when an edge $(v_i, v_j)$ exists within the set $\mathcal{E}$, or $\mathbf{A}_{ij} = 0$, otherwise. The degree matrix is denoted as $\mathbf{D} = \mathrm{diag}(d_1, d_2, \ldots, d_N)$, where each element $d_i = \sum_{j \in \mathcal{V}} \mathbf{A}_{ij}$. The degree-normalized adjacency matrix with self-loops is represented as $\mathbf{A}_{\mathrm{sym}} = \mathbf{D}^{-1/2}(\mathbf{A} + \mathbf{I})\mathbf{D}^{-1/2}$.

**Graph Contrastive Learning.** Given a graph's attributes $\mathbf{X}$ and adjacency matrix $\mathbf{A}$, the objective of GCL is to train an encoder $f(\cdot)$ in a self-supervised fashion. The learned encoder $f(\cdot)$ can generate representations $\mathbf{H} = f(\mathbf{X}, \mathbf{A}), \mathbf{H} \in \mathbb{R}^{N \times K}$, which are both topologies decoupled and dense. These representations can be applied to many downstream tasks.

## 3 Representation Scattering in GCL

We now examine the shared elements that contribute to the effectiveness of popular Graph Contrastive Learning (GCL) frameworks. Upon revisiting three widely used baseline GCL frameworks, we find that they all inherently utilize the mechanism of representation scattering, which plays a crucial role in their success. Here, we formally define Representation Scattering:

**Definition 1.** *(Representation Scattering) In a $d$-dimensional embedding space $\mathbb{R}^d$ comprising $n$ vectors organized into a matrix $\mathbf{V} \in \mathbb{R}^{n \times d}$, consider a subspace $\mathbb{S}^k$ $(1 \leq k \leq d)$ of $\mathbb{R}^d$ and a scatter center $\mathbf{c}$. Representation scattering is a process satisfying two constraints, (i) Center-Away Constraint: Node representations are encouraged to be distant from the scattered center $\mathbf{c}$, and (ii) Uniformity Constraint: Node representations are uniformly distributed over the subspace $\mathbb{S}^k$.*

According to Definition 1, achieving representation scattering requires identifying a scattered center $\mathbf{c}$ within the subspace $\mathbb{S}^k$, and simultaneously satisfying the Center-Away and Uniformity Constraints. We will study its relationship with the popular GCL frameworks in the following section.

### 3.1 DGI-like methods

DGI-like methods generate negative samples through random permutation of nodes. They employ a mutual information discriminator $\mathcal{D}(\cdot)$, which maximizes the mutual information between nodes and their source graphs to train the model [6, 7, 15]. Here, we show that the objective function of DGI is a special case of representation scattering. To facilitate the proof, we present the following assumption:

**Assumption 1.** *(a) The normalized propagation matrix $\tilde{\mathbf{A}}$ is defined as $\tilde{\mathbf{A}} = \mathbf{D}^{-1}\hat{\mathbf{A}}$, where $\hat{\mathbf{A}} = \mathbf{A} + \mathbf{I}$. (b) DGI generates the corrupted graph by randomly shuffling the entities in the feature matrix $\mathbf{X}$, while keeping the adjacency matrix $\mathbf{A}$ unchanged. (c) The original data are class-balanced, i.e., for any classes $k$ and $j$, num$(k) =$ num$(j)$.*

The following results do not strictly require Assumption 1 to be satisfied. Assumption 1 represents a common scenario and serves to simplify the proof. A discussion on the validity of Assumption 1, and proofs of the subsequent results without Assumption 1, can be found in Appendix C.

**Theorem 1.** *At the node level, minimizing the DGI loss is equivalent to maximizing the Jensen-Shannon (JS) divergence between the local semantic distribution in the original graph and its average distribution.*

*Proof.* Let $p_{\mathrm{data}}$ denote the distribution of all nodes in the original graph, characterized by a mean $\mu$ and variance $\sigma^2$. To analyze the local distribution of node $v_i$ after its embedding aggregation by GNN encoders, we define $p_i$ as the distribution of node $v_i$ and its first-order neighbors with mean $\mu_i$ and variance $\sigma_i^2$. We make use of a conclusion in [6] and introduce the following lemma:

**Lemma 2.** *Minimizing the DGI loss, denoted as $\mathcal{L}_{DGI}$, equals to maximizing the Jensen-Shannon (JS) divergence between the distribution of the original graph $\mathcal{G}$ and the corrupted graph $\tilde{\mathcal{G}}$, i.e., $Min(\mathcal{L}_{DGI}) \Leftrightarrow Max(JS(\mathcal{G} \parallel \tilde{\mathcal{G}}))$.*

We discuss Lemma 2 in Appendix D.1. Lemma 2 establishes the relationship between the DGI loss and the distributions of the original and corrupted graphs. To investigate the distribution of representations in original and corrupted graphs, we focus on the case of a single-layer GNN [21] and have the following formulation:

$$\mathbf{H} = \text{GNN}(\mathbf{A}, \mathbf{X}) = \tilde{\mathbf{A}}\mathbf{X}\mathbf{W}', \mathbf{h}_i = \sum_{j \in \mathcal{N}_i} \alpha_{ij}\mathbf{x}_j \cdot \mathbf{W}' = \sum_{j \in \mathcal{N}_i} \frac{1}{d_i}\mathbf{x}_j \cdot \mathbf{W}', \quad (1)$$

where $\mathbf{W}' = \xi(\mathbf{W}), \mathbf{W} \in \mathbb{R}^{D \times K}$ with $\xi$ being an activation function like ReLU [22] for ease of understanding, $\mathcal{N}_i$ denotes the set of first-order neighbors of node $v_i$, inclusive of $v_i$ itself. In Eq. 1, $\forall j \in \mathcal{N}_i, \mathbf{x}_j \sim p_i(\mu_i, \sigma_i^2)$. For node $v_i$, subsequent to GNN message passing, we compute the mean and variance of the aggregated representation $\mathbf{h}_i$ as follows:

$$\mathbb{E}[\mathbf{h}_i] = \mathbb{E}\left[\sum_{j \in \mathcal{N}(i)} \alpha_{ij}\mathbf{x}_j \cdot \mathbf{W}'\right] = \sum_{j \in \mathcal{N}(i)} \alpha_{ij}\mathbb{E}[\mathbf{x}_j] \cdot \mathbf{W}' = \sum_{j \in \mathcal{N}(i)} \alpha_{ij}\mu_i \cdot \mathbf{W}',$$

$$\text{Var}(\mathbf{h}_i) = \text{Var}\left(\sum_{j \in \mathcal{N}(i)} \alpha_{ij}\mathbf{x}_j \cdot \mathbf{W}'\right) = \sum_{j \in \mathcal{N}(i)} \alpha_{ij}\text{Var}(\mathbf{x}_j) \cdot \mathbf{W}'^2 = \sigma_i^2 \sum_{j \in \mathcal{N}(i)} \alpha_{ij}^2 \cdot \mathbf{W}'^2. \quad (2)$$

Given that $\tilde{\mathbf{A}}$ is a normalized propagation matrix with $\sum_{j \in \mathcal{N}(i)} \alpha_{ij} = 1$ for the aggregated representation $\mathbf{h}_i$ in the original graph, the mean of representation $\mathbf{h}_i$'s distribution equals $\mu_i \cdot \mathbf{W}'$. We then analyze the distribution of the aggregated representation $\hat{\mathbf{h}}_i$ in the corrupted graph. Based on Assumption 1, the adjacency matrix $\mathbf{A}$ is unaltered and the feature vector $\tilde{\mathbf{x}}_i$ is selected randomly from the feature matrix $\mathbf{X}$ of the original graph, following the distribution $p_{\text{data}}(\mu, \sigma^2)$. According to Eq.2, the mean of the distribution of $\hat{\mathbf{h}}_i$ is $\mu \cdot \mathbf{W}'$, and the variance is $\sigma^2 \sum_{j \in \mathcal{N}(i)} \alpha_{ij}^2 \cdot \mathbf{W}'^2$. Consequently, we have $\mathbf{h}_i \sim p_i'(\mu_i \cdot \mathbf{W}', \sigma_i^2 \cdot \mathbf{W}'^2)$ and $\hat{\mathbf{h}}_i \sim p_{\text{data}}'(\mu \cdot \mathbf{W}', \sigma^2 \cdot \mathbf{W}'^2)$, which directly prove that maximizing the JS divergence between the original and corrupted graph distributions at the node level maximizes, in effect, the JS divergence between the local semantic distribution of node $v_i$ and the mean distribution of the nodes in the original graph. □

**Corollary 3.** *Taking the mean of the original graph as the center* $\mathbf{c}$*, and the original representation space as a subspace* $\mathbb{S}^k$*, the objective of DGI can be described as follows: within the subspace* $\mathbb{S}^k$*, DGI increases the distance between the nodes of the original graph and its center* $\mathbf{c}$*, achieving the objective of representation scattering.*

Corollary 3 reveals, for the first time, that the primary goal of DGI-like methods is to position node representations away from a central point to encourage a uniform distribution of the nodes. To illustrate these theoretical insights, we performed a visualization experiment. As shown in Figure 1 (a) and (b), both a randomly initialized GNN and a trained single-layer GNN demonstrated that their representations are distanced from the mean of the original graph nodes. Ad-

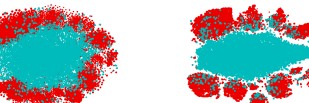 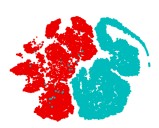

(a) Random Init    (b) GNN-Layer #1    (c) GNN-Layer #2

Figure 1: t-SNE embedding of DGI on Co.CS dataset. The blue points represent the embeddings of the perturbed negative samples, and the red points denote that of positive nodes. As can be seen in Figures (a) and (b) of model random initialization and the layer of the trained encoder, the DGI-like methods essentially maximize the JS divergence between node embedding and embedding mean.

ditionally, Figure 1 (b) and (c) provide clearer evidence that minimizing the $\mathcal{L}_{\text{DGI}}$ is equivalent to maximizing the Jensen-Shannon (JS) divergence between the positive and negative samples, i.e., maximizing the JS divergence between the local semantic distribution in the original graph and its mean distribution. However, the row-wise shuffling mechanism may introduce potential disturbances. In a graph with $n$ nodes, each node $v$ retains its local semantic distribution with a probability of $1/k$ during the shuffling process, where $k$ denotes the number of distinct node types, assuming class balance. Despite this, the non-discriminatory nature of the perturbation means that these unchanged nodes could still be mistakenly classified as negative samples. Consequently, genuine node representations may be incorrectly labeled as negatives, leading to bias in the learning process. The overlapping nodes depicted in Figure 1 intuitively support this perspective.

### 3.2 InfoNCE-based methods

We now show that the mechanism of negative sampling in the InfoNCE-based methods is equivalent to representation scattering.

**Theorem 4.** *Let $\bar{\mathbf{h}} = 1/n \sum_{i=1}^{n} \mathbf{h}_i$, and $sim(\cdot)$ be the cosine similarity function (here, $\mathbf{h}_i$ represents the encoded representation of node $v_i$). For node $v_i$, the lower bound of the InfoNCE loss $\mathcal{L}_{InfoNCE}(\mathbf{h}_i)$ exists: $\mathcal{L}_{InfoNCE}(\mathbf{h}_i) \geq sim(\mathbf{h}_i, \bar{\mathbf{h}}) + \ln(2n)$.*

A detailed proof is given in Appendix D.3. Theorem 4 indicates that when minimizing $\mathcal{L}_{\text{InfoNCE}}(\mathbf{h}_i)$, the similarity between node $v_i$ and mean node $\bar{v}$, i.e., $sim(\mathbf{h}_i, \bar{\mathbf{h}})$ is also minimized. Taking the mean of all nodes as the scattered center $\mathbf{c}$, and the hypersphere as the subspace $\mathbb{S}^k$, the mechanism of negative sampling is equivalent to representation scattering. However, the InfoNCE loss function is inefficient for representation scattering as it needs to compute and reduce the similarity of each negative pair. Moreover, it indiscriminately treats all negative samples, ignoring the distinctions among them, which leads to inappropriate scattering of negative samples and potential bias from false negatives. Consequently, many recent methods have incurred additional computational overhead by manually and intuitively defining positive and negative samples [12, 23].

### 3.3 BGRL-like methods

All BGRL-like methods incorporate a key component: Batch Normalization (BN). For a feature vector $\mathbf{x}_i$ and its corresponding batch statistics, mean $\mu$ and variance $\sigma^2$, the BN is applied as follows: $\text{BN}(\mathbf{x}_i) = \gamma(\mathbf{x}_i - \mu)/\sqrt{\sigma^2 + \epsilon} + \beta$, where $\gamma$ and $\beta$ are learnable parameters that scale and shift the normalized value, and $\epsilon$ is a small constant added for numerical stability.

**Theorem 5.** *The process of data normalization by batch normalization can be seen as a special case of representation scattering.*

A detailed proof can be found in Appendix D.2. To empirically assess the impact of batch normalization on BGRL, we conducted experiments comparing both BGRL and BGRL w/o BN across four benchmark datasets. The results, shown in Figure 2, reveal a significant drop in accuracy for BGRL across all datasets without batch normalization, highlighting its critical role in the bootstrapping framework. While BGRL incorporates representation scattering through batch normalization, its training process lacks explicit guidance or a dedicated mechanism to efficiently manage this scattering. The absence of direct supervision during the representation scattering phase can lead to an unoptimized distribution of representation within the embedding space, resulting in suboptimal performance.

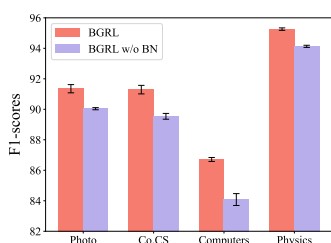

Figure 2: The impact of Batch Normalization in BGRL.

In summary, our analysis of the three mainstream GCL frameworks theoretically proved that they all inherently utilize representation scattering but also fail to fully utilize this effective mechanism. This analytic result has motivated us to design a more effective representation scattering method tailored for learning on graphs.

## 4 Methodology

The proceeding sections have revealed the importance of representation scattering in GCLs. Based on the findings, we have designed a novel method, namely Scattering Graph Representation Learning, short-handed as SGRL (Figure 3). We introduce the components of SGRL and provide discussion in the following sub-sections.

### 4.1 Representation Scattering Mechanism (RSM)

To address the shortcomings in the application of representation scattering within the three mainstream graph contrastive learning frameworks, we design RSM to explicitly guide the target encoder in learning scattered representations. Following Definition 1, we introduce a subspace $\mathbb{S}^k$ and a scattered

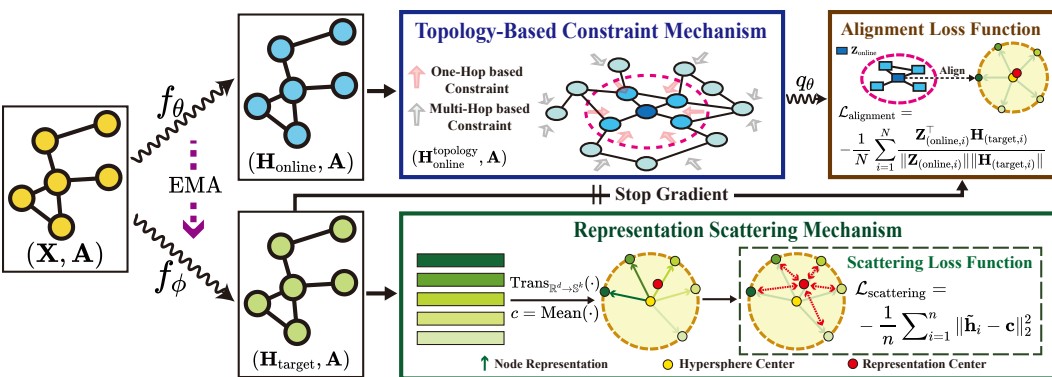

Figure 3: The overview of SGRL. Consider a graph $\mathcal{G}$ processed using two distinct encoders (online encoder and target encoder): $f_\theta(\cdot)$ with parameters $\theta$ and $f_\phi(\cdot)$ with $\phi$, aimed at generating node representations $\mathbf{H}_{\text{online}}$ and $\mathbf{H}_{\text{target}}$, respectively. For $\mathbf{H}_{\text{target}}$, the mean representation of all nodes is calculated to serve as the scattered center $\mathbf{c}$. The parameters of $f_\phi(\cdot)$ are updated via RSM that encourages node representations to diverge from $\mathbf{c}$. $\mathbf{H}_{\text{online}}$ is processed through TCM to incorporate topology information, resulting in $\mathbf{H}_{\text{online}}^{\text{topology}}$. Subsequently, $\mathbf{H}_{\text{online}}^{\text{topology}}$ is embedded through a predictor $q_\theta$ to predict $\mathbf{H}_{\text{target}}$, and the parameters in $f_\theta(\cdot)$ is updated through back-propagation while stopping the gradient of $f_\phi(\cdot)$. Both channels are trained simultaneously. At the end of each epoch, we employ an Exponential Moving Average (EMA) to update parameters $\phi$. Finally, the representations generated by $f_\theta(\cdot)$ are employed across various downstream tasks.

center $\mathbf{c}$ to effectively perform representation scattering. For the subspace $\mathbb{S}^k$, a transformation function $\text{Trans}(\cdot)$ is introduced to transform representations from the original space $\mathbb{R}^d$ into $\mathbb{S}^k$. Specifically, we apply $\ell_2$ normalization to each row vector $\mathbf{h}_i$ in the matrix $\mathbf{H}_{\text{target}}$:

$$\tilde{\mathbf{h}}_i = \text{Trans}_{\mathbb{R}^d \to \mathbb{S}^k}(\mathbf{h}_i) = \frac{\mathbf{h}_i}{\text{Max}(\|\mathbf{h}_i\|_2, \varepsilon)}, \quad \mathbb{S}^k = \{\tilde{\mathbf{h}}_i : \|\tilde{\mathbf{h}}_i\|_2 = 1\}, \tag{3}$$

where $\mathbf{h}_i$ is representation for node $v_i \in \mathcal{V}$, generated by the target encoder, $\|\tilde{\mathbf{h}}_i\|_2 = (\sum_{j=1}^k \tilde{\mathbf{h}}_{ij}^2)^{\frac{1}{2}}$, and $\epsilon$ is a small value to avoid division by zero. As defined by Eq. 3, the representations of all nodes are distributed on a hypersphere $\mathbb{S}^k$. This mapping prevents arbitrary scattering of representations in the space, avoiding instability and optimization difficulties during training.

Next, we define the scattered center $\mathbf{c}$ and introduce a representation scattering loss function $\mathcal{L}_{\text{scattering}}$ in $\mathbb{S}^k$ to push node representations away from the center $\mathbf{c}$, as formulated as follows:

$$\mathcal{L}_{\text{scattering}} = -\frac{1}{n} \sum_{i=1}^n \|\tilde{\mathbf{h}}_i - \mathbf{c}\|_2^2, \quad \mathbf{c} = \frac{1}{n} \sum_{i=1}^n \tilde{\mathbf{h}}_i. \tag{4}$$

Through Eq. 4, SGRL achieves uniformity of representations globally across the entire dataset, without emphasizing local uniformity. Specifically, RSM enables representations of different semantics to be globally scattered across the hypersphere while accommodating representations of the same semantics to aggregate locally.

**Discussion of RSM.** We now provide a theoretical analysis demonstrating that the proposed representation scattering mechanism outperforms the three graph contrastive learning frameworks. RSM achieves representation scattering more effectively by encouraging distances between node representations and the scattered center, eliminating the dependence on manually designed negative samples. Most traditional methods [7, 8, 9, 15] rely on negative samples to indirectly promote representation scattering, which is inefficient and introduces biases. DGI-like methods, as discussed in Section 3.1, aim to maximize the Jensen-Shannon (JS) divergence between the distribution of the original graph and its mean distribution. Based on this, generating additional noise graph is inefficient. Moreover, some negative samples generated through random shuffling may align with the distribution of positive samples, resulting in false negative samples that hinder model training. InfoNCE-based methods consider all nodes as negative samples, except the matching ones in two augmented views. While pushing nodes away from each other ensures the discriminability of each node, it also results in significant computational overhead. Moreover, due to the potential conflict between the encoder's message-passing mechanism and the InfoNCE loss function, many negative samples can not effectively distance themselves from each other [24]. Therefore, by employing a center-away strategy,

RSM effectively reduces the additional computational overhead and mitigates the biases caused by manually designed negative samples.

## 4.2 Topology-based Constraint Mechanism(TCM)

After obtaining the scattered representations $\mathbf{H}_{\text{target}} = f_\phi(\mathbf{A}, \mathbf{X})$ through the target encoder, it is necessary to consider the differences in the degree of scattering of different node representations.

Consider the interconnected nature of graphs, the representations of two nodes connected topologically should be closer in space $\mathbb{S}^k$. Specifically, $\forall v_i, v_j \in V$, $\mathbf{h}_i, \mathbf{h}_j \in \mathbb{S}^k$, given a threshold $d$: $\|\mathbf{h}_i - \mathbf{h}_j\|_2^2 < d$, if $(i, j) \in \mathcal{V}$ and $\|\mathbf{h}_i - \mathbf{h}_j\|_2^2 > d$, if $(i, j) \notin \mathcal{V}$. Intuitively, to this end, a simple way is to replace the individually scattered representations with the aggregated representations $\mathbf{H}_{\text{target}}^{\text{topology}}$ from first-order neighbors: $\mathbf{H}_{\text{target}}^{\text{topology}} = \mathbf{A}\mathbf{H}_{\text{target}}$. However, attempting to consider the topology information and achieve representation scattering through the same encoder may lead to conflicts.

To address this issue, we propose a Topology-based Constraint Mechanism (TCM). Specifically, we separate the process of constraint from the process of scattering by letting the online encoder generate topologically aggregated representations instead of the target. The online encoder enhances its representations by summing the original representations $\mathbf{H}_{\text{online}}$ with the topologically aggregated representations of its k-order neighbors $\hat{\mathbf{A}}^k\mathbf{H}_{\text{online}}$, which can be described as:

$$\mathbf{H}_{\text{online}}^{\text{topology}} = \hat{\mathbf{A}}^k\mathbf{H}_{\text{online}} + \mathbf{H}_{\text{online}}, \tag{5}$$

where k represents the order of neighbors and $\hat{\mathbf{A}} = \mathbf{A} + \mathbf{I}$ is the adjacency matrix with self-loops. By separating scattering and constraints, SGRL can effectively achieve a balance between representation scattering and topology aggregation adaptively, rather than setting the scattering distance empirically. Next, the topology representations $\mathbf{H}_{\text{online}}^{\text{topology}}$ are fed into a predictor $q_\theta(\cdot)$ to generate the predicted representations $\mathbf{Z}_{\text{online}} = q_\theta(\mathbf{H}_{\text{online}}^{\text{topology}})$. Our objective is to align the predicted topology representations $\mathbf{Z}_{\text{online}}$ closely to the scattered representations $\mathbf{H}_{\text{target}}$, enhancing the model's effectiveness in capturing the essential semantic details of the graph. Based on this, the alignment loss $\mathcal{L}_{\text{alignment}}$ is defined as follows:

$$\mathcal{L}_{\text{alignment}} = -\frac{1}{N} \sum_{i=1}^{N} \frac{\mathbf{Z}_{(\text{online},i)}^\top \mathbf{H}_{(\text{target},i)}}{\|\mathbf{Z}_{(\text{online},i)}\|\|\mathbf{H}_{(\text{target},i)}\|}, \tag{6}$$

where $\mathbf{Z}_{\text{online}}$ and $\mathbf{H}_{\text{target}}$ represent the predicted and scattered representations, respectively. During this process, the online encoder's parameters $\theta$ are updated and the target encoder's parameters $\phi$ stop gradient propagation. Compared to directly aligning constrained and scattered representations, this predictive objective can act as a buffer, allowing the online encoder to adaptively learn scattered representations and topology information. Furthermore, to make the target encoder consider topological semantic information into the process of representation scattering, instead of solely focusing on scattering, we employ an Exponential Moving Average at the end of each training epoch:

$$\phi \leftarrow \tau\phi + (1 - \tau)\theta, \tag{7}$$

where $\tau$ is a target decay rate and $\tau \in [0, 1]$. Eq. 7 effectively mitigates the adversarial interactions between RSM and TCM, while also facilitating the integration of topology information into the representation scattering process. Moreover, RSM enhances the discriminability of representations, while TCM incorporates topology information into the representations. The interaction between these two mechanisms effectively mimics the role of data augmentation, i.e., train encoders to learn the invariance of data to perturbations. Consequently, SGRL obviates the need for explicitly designing data augmentation strategies, which leads to additional computational overhead and heavy reliance on the choice of augmentation techniques.

## 5 Experiment

We evaluated SGRL on the five of the most widely used benchmark datasets, including Amazon-Photo (Photo) and Amazon-Computers (Computers) [25], WikiCS [26], Coauthor-CS (Co.CS) and Coauthor-Physics (Co.Physics) [27]. Detailed information of these datasets is in Appendix B.2. We compared SGRL with four types of methods: (1) Three mainstream baselines: GRACE [8],

DGI [7], and BGRL [10]. (2)Six recently advanced algorithms: GCA [9], ProGCL [12], AFGRL [13], iGCL [28], GBT[29], MVGRL [15]. (3)Two classic graph representation learning methods: Node2vec [30] and Deepwalk [31]. (4)The semi-supervised training baseline GCN [18]. We utilized the representations generated by the online encoder for downstream tasks. For node classification, we followed the evaluate scheme from [10]. We trained a simple linear model using only the representations from a logistic regression loss with an $\ell_2$ regularization and no backpropagation of any gradient to the graph encoder network. Specifically, we trained the downstream classifier using $10\%$ of the data and tested the classifier on the remaining $90\%$. We ran SGRL 20 times and report here the average and standard deviation of the F1-score. For node clustering, we adopted the evaluation method from [13]. The testing was conducted on the learned representations at each epoch, and the best performance is reported below. More details of the experiments can be found in Appendix B.

Table 1: Performance on node classification. OOM signifies out-of-memory on 24GB RTX 3090. $X, A, Y$ denote the node attributes, adjacency matrix, and labels in the datasets. Optimal results are shown in bold.

| Method | Available Data | WikiCS | Computers | Photo | Co.CS | Co.Physics |
|---|---|---|---|---|---|---|
| Raw Features | $X$ | 71.98 ± 0.00 | 73.81 ± 0.00 | 78.53 ± 0.00 | 90.37 ± 0.00 | 93.58 ± 0.00 |
| Node2vec | $A$ | 71.79 ± 0.05 | 84.39 ± 0.08 | 89.67 ± 0.12 | 85.08 ± 0.03 | 91.19 ± 0.04 |
| DeepWalk | $A$ | 74.35 ± 0.06 | 85.68 ± 0.06 | 89.44 ± 0.11 | 84.61 ± 0.22 | 91.77 ± 0.15 |
| GRACE | $X, A$ | 77.97 ± 0.63 | 86.50 ± 0.33 | 92.46 ± 0.18 | 92.17 ± 0.04 | OOM |
| DGI | $X, A$ | 75.35 ± 0.14 | 83.95 ± 0.47 | 91.61 ± 0.22 | 92.15 ± 0.63 | 94.51 ± 0.52 |
| BGRL | $X, A$ | 76.86 ± 0.74 | 89.69 ± 0.37 | 93.07 ± 0.38 | 92.59 ± 0.14 | 95.48 ± 0.08 |
| GBT | $X, A$ | 76.65 ± 0.62 | 88.14 ± 0.33 | 92.63 ± 0.44 | 92.95 ± 0.17 | 95.07 ± 0.17 |
| MVGRL | $X, A$ | 77.52 ± 0.08 | 87.52 ± 0.11 | 91.74 ± 0.07 | 92.11 ± 0.12 | 95.33 ± 0.03 |
| GCA | $X, A$ | 77.94 ± 0.67 | 87.32 ± 0.50 | 92.39 ± 0.33 | 92.84 ± 0.15 | OOM |
| ProGCL | $X, A$ | 78.45 ± 0.04 | 89.55 ± 0.16 | 93.64 ± 0.13 | 93.67 ± 0.12 | OOM |
| AFGRL | $X, A$ | 77.62 ± 0.49 | 89.88 ± 0.33 | 93.22 ± 0.28 | 93.27 ± 0.17 | 95.69 ± 0.10 |
| iGCL | $X, A$ | 78.83 ± 0.08 | 89.41 ± 0.06 | 93.02 ± 0.06 | 93.52 ± 0.04 | 94.77 ± 0.20 |
| **SGRL(Ours)** | $X, A$ | **79.40 ± 0.10** | **90.23 ± 0.03** | **93.95 ± 0.03** | **94.15 ± 0.04** | **96.23 ± 0.01** |
| Supervised GCN | $X, A, Y$ | 77.19 ± 0.12 | 86.51 ± 0.54 | 92.42 ± 0.22 | 93.03 ± 0.31 | 95.65 ± 0.16 |

## 5.1 Performance Analysis

**Overall evaluation.** Table 1 presents the averages and standard deviations of the F1-scores for all methods on the node classification task. Some statistics for the existing methods are reported from either their original papers or [13]. As shown in the table, the proposed SGRL exhibits superior performance across all five datasets tested, achieving the highest accuracy. Compared to the three baselines, BGRL, GRACE, and DGI, our new method outperforms them by 1.23%, 2.14%, and 3.26%, respectively, showing SGRL's effectiveness in more effectively exploiting representation scattering. Moreover, we observe that SGRL outperforms methods like GCA, AFGRL and iGCL, which focus on improving upon data augmentation.

Table 2: Performance on Clustering in terms of NMI and homogeneity. Optimal results are shown in bold and suboptimal results are underlined.

| | | GRACE | DGI | BGRL | **SGRL** |
|---|---|---|---|---|---|
| WikiCS | NMI | 0.4282 | **0.4312** | 0.3969 | 0.4188 |
| | Hom. | 0.4423 | **0.4498** | 0.4156 | 0.4369 |
| Amazon-Computers | NMI | 0.4793 | 0.4630 | 0.5364 | **0.5380** |
| | Hom. | 0.5222 | 0.4836 | **0.5869** | 0.5705 |
| Amazon-Photo | NMI | 0.6513 | 0.5487 | **0.6841** | 0.6788 |
| | Hom. | 0.6657 | 0.5557 | **0.7004** | 0.6786 |
| Co-CS | NMI | 0.7562 | 0.7162 | 0.7732 | **0.7961** |
| | Hom. | 0.7909 | 0.7428 | 0.8041 | **0.8216** |
| Co-Physics | NMI | OOM | 0.6540 | 0.5568 | **0.7232** |
| | Hom. | OOM | 0.6868 | 0.6018 | **0.7366** |

This demonstrates the validity of employing a graph contrastive framework based on representation scattering and topology aggregation, instead of using data augmentation. SGRL also outperforms the advanced negative sampling method, ProGCL. We attribute this to SGRL's representation scattering, which does not explicitly define negative samples, achieving better performance than manually defined negative samples. When evaluated on node clustering tasks using the scheme in [13], it can be observed that SGRL achieves the best or second-best accuracy on most datasets. Although SGRL attempts to make node representations scattered, which seems unsuitable for clustering tasks, it still manages to deliver competitive performance. This demonstrates that TCM can preserve the topology information of the original graph in the process of representation scattering, ensuring that while

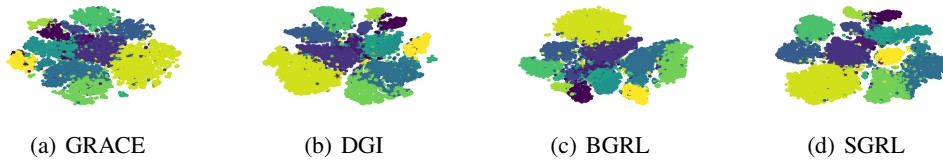

|              |              |              |              |
|:------------:|:------------:|:------------:|:------------:|
| (a) GRACE    | (b) DGI      | (c) BGRL     | (d) SGRL     |

Figure 4: t-SNE embeddings of nodes in CS dataset.

performing representation scattering, it maintains the local structure and global semantic consistency of the nodes.

**Visualization.** To more intuitively show the advantages of the representations learned by SGRL, we employ t-SNE [32] for visualizing the learned representations from the Co-CS dataset. Each point represents a node, with the color indicating the node's label. As shown in Figure 4, SGRL shows clearer inter-class boundaries compared to the other three methods, indicating that RSM achieves superior scattering. Moreover, we observe better intra-class clustering in SGRL. This highlights that SGRL, which does not require manually defined negative samples, prevents the inappropriate distancing of intra-class nodes. It achieves both global scattering and local semantic aggregation, demonstrating the effectiveness of adaptive scattering methods based on topology constraints.

Table 3: Ablation study on node classification. Optimal results are shown in bold.

| Variant | WikiCS | Amazon-Computers | Amazon-Photo | Co-CS | Co-Physics |
|---|---|---|---|---|---|
| SGRL w/o RSM and TCM | 76.86 ± 0.74 | 89.69 ± 0.37 | 93.07 ± 0.38 | 92.59 ± 0.14 | 95.48 ± 0.08 |
| SGRL w/o TCM | 78.55 ± 0.08 | 89.54 ± 0.10 | 93.58 ± 0.05 | 94.08 ± 0.03 | 96.19 ± 0.04 |
| SGRL w/o EMA | 79.36 ± 0.08 | 90.03 ± 0.07 | 93.92 ± 0.02 | 93.89 ± 0.07 | 96.16 ± 0.07 |
| SGRL (Ours) | **79.40 ± 0.13** | **90.23 ± 0.03** | **93.95 ± 0.03** | **94.15 ± 0.04** | **96.23 ± 0.01** |

## 5.2 Model Analysis

**Hyperparamter Analysis.** In this subsection, we investigate the sensitivity of the hyperparameter k in SGRL, as shown in Eq. 5. The parameter k represents the order of neighbors aggregated, which directly influences the strength of the topology constraints. In our experiments, we adjust k in the range $0, 1, 2, \cdots, 7$ to evaluate the impact of different constraint strengths on SGRL. The results are shown in in Figure 5: when k = 0, i.e., "SGRL w/o TCM", the lack of topology constraints leads to a decrease in model performance. As k increases, the constraint ability of TCM gradually strengthens, and the model performance exhibits a unimodal shape with respect to changes in k. This highlights that weak topology cannot preserve adequate topological information for adaptive representation scattering, whereas strong topology may excessively restrict the scattering of representations, which aligns with the perspectives proposed in Sections 3 and 4.

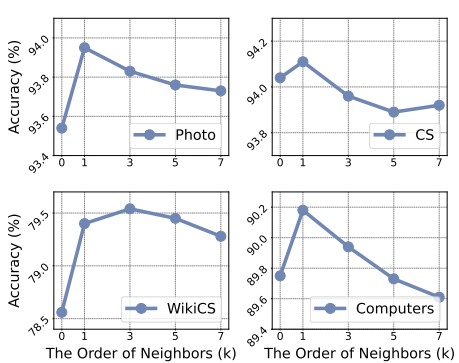

Figure 5: Hyperparameter Analysis on k

**Ablation Studies.** To verify the effectiveness of each component of SGRL, we conducted ablation studies on five datasets. As shown in Table 3, "SGRL w/o TCM and TCM" demonstrates significant performance improvements on four datasets with the addition of RSM alone, i.e., "SGRL w/o TCM", confirming its effectiveness. We observe that there is a slight performance decrease on the Computers dataset, which we attribute to over-scattering resulting from the absence of constraints during representation scattering. Additionally, a comparison between SGRL and "SGRL w/o TCM" reveals that SGRL achieves improved accuracy across all five datasets, particularly on the Computers dataset. This indicates that the TCM effectively constrain scattering, achieving an adaptive representation

scattering, verifying our view in Section 4.2. Finally, we performed an ablation study on EMA. Given the potential adversarial relationship between RSM and TCM, we employed EMA to balance this effect. The experimental results in Table 3 highlight the necessity of incorporating EMA to mitigate the adversarial interaction between RSM and TCM.

## 6 Conclusion

In this paper, we made two significant contributions to Graph Contrastive Learning (GCL), an actively researched subject with numerous applications across diverse domains. First, through a comprehensive analysis of the three popular GCL frameworks, we discovered a common latent mechanism – representation scattering – that underlies these distinct contrastive methods. This discovery highlights an essential feature of GCL and unifies these seemingly disparate frameworks under the concept of representation scattering. Despite their popularity, the existing methods have not fully leveraged this mechanism, leaving the potential of representation scattering largely untaped. However, applying representation scattering directly to GCL poses technical challenges. Our second contribution addressed this issue by introducing the Representation Scattering Mechanism (RSM) and the Topology-based Constraint Mechanism (TCM), which we integrate into a novel GCL approach named Scattering Graph Representation Learning (SGRL). In SGRL, we effectively balance the adversarial relationship between RSM and TCM using an Exponential Moving Average (EMA) strategy. Extensive experimental results on benchmark datasets validate the effectiveness of our proposed method. Future work will further explore the broader implications of representation scattering beyond GCL, discussed in Appendices F and G.

## Acknowledgement

This work was supported by the National Natural Science Foundation of China [No. 62422210, No. U22B2036, No. 62276187], the National Science Fund for Distinguished Young Scholarship (No. 62025602), the XPLORER PRIZE, the Hong Kong RGC theme-based Strategic Target Grant (STG1/M-501/23-N), the Hong Kong Global STEM Professorship Scheme, and the Hong Kong Jockey Club Charity Trust.

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

# A  Discussion of Uniformity

In visual contrastive learning, uniformity is recognized as a critical factor contributing to the effectiveness of contrastive methods. In [17], uniformity is defined as the condition where feature vectors should be roughly uniformly distributed on the unit hypersphere, preserving as much information of the data as possible. This concept is based on the InfoNCE loss. Due to the similarities in implementation mechanisms between the SimCLR framework and the Deep InfoMax (DIM) framework, DIM can be considered a special case of SimCLR when the number of negative samples within a batch is limited to one. Consequently, this concept can be seen as a key factor contributing to the success of most visual contrastive learning methods.

However, this concept fails to adequately unify the frameworks within graph contrastive learning, particularly for DGI framework. In DGI, all nodes of the original graph are treated as a single distribution, whereas the distribution generated from a randomly shuffling noise graph is treated as another. The encoder is trained by maximizing the Jensen-Shannon (JS) divergence between these two distributions. First, the representations in DGI do not conform to a distribution on the unit hypersphere. Moreover, considering all nodes of the original graph as one distribution more closely aligns with a process of aggregation rather than facilitating a uniform distribution of nodes throughout the space.

Furthermore, while the negative sampling mechanism of InfoNCE-based methods implements uniformity, the interconnected nature of graphs can lead to conflicts during message passing within the encoder, which hinders the effective distancing of negative samples [24]. Therefore, we conducted a comprehensive theoretical analysis of three mainstream graph contrastive frameworks (Section 3) and introduced the concept of representation scattering. Specifically, the center-away constraint within representation scattering provides a more detailed elucidation of the operational mechanisms underlying the DGI framework than uniformity. Moreover, this constraint effectively mitigates the previously discussed conflicts, thereby facilitating the integration of representations and topological information and enabling adaptive representation scattering.

# B  Experiment Details

## B.1  Hyperparameters Settings

Table 4: Detailed hyperparameters of SGRL.

| Dataset | Hidden dim | online learning rate | target learning rate | Training epochs | Activation | momentum |
|---|---|---|---|---|---|---|
| WikiCS | 1024 | 0.00001 | 0.00001 | 500 | PReLU | 0.99 |
| Amazon-Computers | 1024 | 0.001 | 0.001 | 700 | PReLU | 0.99 |
| Amazon-Photo | 1024 | 0.001 | 0.001 | 700 | PReLU | 0.99 |
| Coauthor-CS | 1024 | 0.001 | 0.001 | 700 | PReLU | 0.99 |
| Coauthor-Physics | 1024 | 0.0001 | 0.00001 | 1000 | PReLU | 0.99 |

## B.2  Datasets

We use Wiki-CS, Amazon-Computers (Computers), Amazon-Photo (Photo), Coauthor-CS and Coauthor-Physics to evaluate the performance of SGRL. The detailed statistics of all the used datasets are in Table 5

Table 5: Statistics of datasets used in this paper.

| Dataset | Type | Nodes | Edges | Attributes | Classes |
|---|---|---|---|---|---|
| Wiki-CS | reference | 11,701 | 216,123 | 300 | 10 |
| Amazon-Computers | co-purchase | 13,381 | 245,778 | 767 | 10 |
| Amazon-Photo | co-purchase | 7,487 | 119,043 | 745 | 8 |
| Coauthor-CS | co-author | 18,333 | 81,894 | 6,805 | 15 |
| Coauthor-Physics | co-author | 34,493 | 247,962 | 8,415 | 5 |

**Wiki-CS** [26] is a directed graph dataset sourced from Wikipedia. It comprises nodes that represent articles in the field of computer science, with edges corresponding to the hyperlinks interconnecting

these articles. Each article is categorized into one of 10 associated subfields. The attributes of the nodes are derived as the average of the text embeddings for the respective articles, making it the sole dataset characterized by dense attributes.

**Amazon-Computers and Amazon-Photo** [25] are networks that map out the co-purchase relationships among products on the Amazon platform. In these networks, nodes represent products, and edges are established between products that are frequently bought together. Products within each network are categorized into 10 and 8 distinct classes respectively, according to their product categories, and node features are represented using a bag-of-words model from the product reviews.

**Coauthor-CS and Coauthor-Physics** [27] represent two academic networks based on the Microsoft Academic Graph, illustrating the co-authorship links among academics. In these networks, nodes correspond to individual authors, while edges reflect collaborative authorship between them. Authors in each network are categorized into 15 and 5 research fields respectively, with node features encapsulated as a bag-of-words representation derived from the keywords of their publications.

### B.3 Environment Configurations

All experiments were carried out on an NVIDIA GeForce GTX 3090 GPU, which comes equipped with 24GB of memory. For model development, we utilized PyTorch version 1.13.1 [33], along with PyTorch Geometric version 2.3.0 [34], which also served as the source for all the datasets used in our study. The detailed hyperparameters for node classification and clustering are shown in Table 4

### B.4 Implementation Details

Our method employs GCN[18] as the encoder. The encoding process can be described as follows:

$$\mathbf{H}_k^{(l)} = \text{GCN}^{(l)}(\mathbf{H}_{k-1}^{(l)}, \mathbf{A}), \tag{8}$$

where $\mathbf{H}_k^{(l)}$ represents the node embedding matrix at the $l$-th layer during the $k$-th epoch of encoding and $\mathbf{H}_0^{(l)}$ is initialized by graph feature matrix $\mathbf{X}$,and $\mathbf{A}$ is the adjacency matrix without self-loops. In addition, the encoder structure is defined as:

$$\text{GCN}^{(l)}(\mathbf{H}_k^{(l)}, \mathbf{A}) = \sigma(\hat{\mathbf{D}}^{-1/2}\hat{\mathbf{A}}\hat{\mathbf{D}}^{-1/2}\mathbf{H}_{k-1}^{(l)}\mathbf{W}_{k-1}^{(l)}), \tag{9}$$

where $\hat{\mathbf{A}} = \mathbf{A} + \mathbf{I}$ is the adjacency matrix with self-loops, $\sigma(\cdot)$ is the PReLU activation function, $\hat{\mathbf{D}}$ is the degree matrix of $\hat{\mathbf{A}}$, and $\mathbf{W}^{(l)}$ is the trainable weight matrix for the $l$-th layer. In our experiments, we set $l = 1$.

## C Detailed Discussion on Assumption 1

**Assumption 1.(a)** In assumption 1.(a), we define the normalized propagation matrix as $\tilde{A} = D^{-1}A$. When the encoder is GCN [18], assumption 1.(a) holds. Next, we will discuss whether assumption 1.(a) holds in Corollary 3 when the encoder is GAT [19]. The message-passing formula for GAT is similar to that of GCN. For node $v_i$, $\mathbf{h}_i = \text{GAT}(A, X) = \sum_{\text{neighbor } j} \alpha_{ij}\mathbf{x}_{ij} \cdot \mathbf{W}$, but the method for obtaining the propagation matrix is different:

$$\alpha_{ij} = \frac{\exp(\text{LeakyReLU}(\mathbf{a}^T[\mathbf{W}\mathbf{x}_i\|\mathbf{W}\mathbf{x}_j]))}{\sum_{k\in\mathcal{N}(i)} \exp(\text{LeakyReLU}(\mathbf{a}^T[\mathbf{W}\mathbf{x}_i\|\mathbf{W}\mathbf{x}_k]))}, \tag{10}$$

where $\mathbf{a}$ is the weight vector used to compute the attention scores, $\|\cdot\|$ denotes the concatenation operation, $\mathbf{W}$ is the weight matrix, and LeakyReLU is the activation function. In the original graph, node $v_i$ and its first-order neighbors follow a distribution with mean $\mu_i$ and variance $\sigma_i^2$, i.e., $\forall j \in \mathcal{N}_i, x_j \sim p_i(\mu_i, \sigma_i^2)$. Subsequent to GAT aggregation, the representation of node $v_i$, i.e., $\mathbf{h}_i$ follows a local semantic distribution with a mean given by $\mathbb{E}\left(\sum \alpha_{ij}x_{ij}\right) = \sum \alpha_{ij}\mu_i$ and a variance of $\sigma_i^2 \sum \alpha_{ij}^2$. This is consistent with the conclusion in Section 3.1. In the perturbed graph, the method for calculating the aggregated distribution differs from that of GCN. For GCN, the propagation matrix $\alpha_{ij} = 1/d_i$ is independent of the distribution of node $v_i$ and its neighbors. In contrast, for GAT, as shown in Eq. 10, $\alpha_{ij}$ reflects the correlation between node $v_i$ and its neighbor $v_j$. Therefore, in

the perturbed graph, there are two distributions present among node $v_i$ and its first-order neighbors: the local semantic distribution $p_i$ and the original data distribution $p_{data}$. Let $a = \sum \alpha_{ik}$, where $x_k \sim p_i(\mu_i, \sigma_i^2)$. Let $b = \sum \alpha_{ik'}$, where $x_{k'} \sim p_{data}(\mu, \sigma^2)$. After aggregation through GAT, the mean of the distribution that node $v_i$ follows is $a \cdot \mu_i + b \cdot \mu$. In this case, minimizing the DGI loss is equivalent to maximizing the local semantic distribution of node $v_i$ with a mean of $a \cdot \mu_i + b \cdot \mu$ and a variance of $\sigma_i^2 \sum \alpha_{ik}^2 + \sigma^2 \sum \alpha_{ik'}^2$.

In our problem, we aim to find a subspace $\mathbb{S}^k$ and a scattered center $\mathbf{c}$ to achieve representation scattering, and this goal can be accomplished with either GCN or GAT. To this end, Corollary 3 also holds when the encoder is GAT.

**Assumption 1.(b)** For assumption 1.(b), there are two ways to generate a corrupted graph in DGI [7]: one is by randomly shuffling the feature matrix $\mathbf{X}$, and the other is by keeping $\mathbf{X}$ unchanged and shuffling the adjacency matrix $\mathbf{A}$. Here, we discuss whether Theorem 1 still holds when the assumption is the latter.

We have already demonstrated in Section 3.1 that, in the original graph $\mathcal{G}$, the aggregated features of node $v_i$ follow a distribution with mean $\mu_i$ and variance $\sigma_i^2 \sum_{j \in \mathcal{N}(i)} \alpha_{ij}^2$. In the corrupted graph $\tilde{G}$, randomly shuffling the adjacency matrix $\mathbf{A}$ is equivalent to randomly re-wiring $n$ nodes, where $\sum_{i=1}^n \sum_{j=1}^n \mathbf{A}_{ij} = \sum_{i=1}^n \sum_{j=1}^n \hat{\mathbf{A}}_{ij}$. The first-order neighbors of node $v_i$ still derive their features from the original graph distribution, i.e., $\hat{x}_{ij} \sim p_{\text{data}}(\mu, \sigma^2)$, and in the corrupted graph, the new topology still satisfies $\sum(\alpha_{ij}) = 1$. Therefore, the aggregated representation follows a distribution with mean $\mu$ and variance $\sigma^2 \sum_{j \in \mathcal{N}(i)} \alpha_{ij}^2$, which is consistent with the conclusion derived from assumption (b).

**Assumption 1.(c)** In our theoretical analysis, we assume class balance, which is common in many theoretical proofs [7]. When the classes are imbalanced, the conclusion is similar to what we explained in assumption 1. (a): the scattered center changes from the mean to $a \cdot \mu_i + b \cdot \mu$, i.e., it leans towards the classes with a larger number of instances. This often occurs in long-tail distribution problems. We will further explore this aspect in our future work.

# D   Proof

## D.1   Proof of Lemma 2

*Proof.* Here, we rewrite the proof from work [6] for ease of understanding Lemma 2. We replace the summary vector $\mathbf{s}$ with an all-ones vector. By removing the weight parameters $\mathbf{W}$ in the discriminator $\mathcal{D}$, we obtain:

$$
\begin{aligned}
\mathcal{L}_{DGI} &= \frac{1}{2N} \left( \sum_{i=1}^N \log \mathcal{D}(\mathbf{h}_i, \mathbf{s}) + \log(1 - \mathcal{D}(\tilde{\mathbf{h}}_i, \mathbf{s})) \right) \\
&= \frac{1}{2N} \left( \sum_{i=1}^N \log(\mathbf{h}_i \cdot \mathbf{s}) + \log(1 - \tilde{\mathbf{h}}_i \cdot \mathbf{s}) \right) \\
&= \frac{1}{2N} \left( \sum_{i=1}^N \log(\text{sum}(\mathbf{h}_i)) + \log(1 - \text{sum}(\tilde{\mathbf{h}}_i)) \right),
\end{aligned}
\tag{11}
$$

where $\text{sum}(\cdot)$ is the summation function for every dimension in vector $\mathbf{h}_i$. As described by Eq. 11, minimizing DGI loss can be viewed as a binary classification problem distinguishing between $\text{sum}(h_i)$ from the original graph and $\text{sum}(\tilde{h}_i)$ from the perturbed graph. Therefore, minimizing DGI loss is equivalent to maximizing the JS divergence between the original and perturbed graphs. $\square$

## D.2   Proof of Theorem 5

*Proof.* Consider the output of a single-layer Graph Neural Network (GNN) as $\mathbf{H} \in \mathbb{R}^{N \times K}$, where $N$ represents the batch size and $K$ denotes the dimension of representations. Each column $\mathbf{H}_j$ of $\mathbf{H}$ represents all samples for a particular feature, with mean $\mu_j$ and standard deviation $\sigma_j$.

Batch Normalization (BN) processes each feature dimension $j$ independently by calculating the mean $\mu_j$ and variance $\sigma_j^2$, and then applying the following transformation:

$$\mathbf{H}_j' = \text{BN}(\mathbf{H}_j) = \frac{\mathbf{H}_j - \mu_j}{\sqrt{\sigma_j^2 + \epsilon}}, \quad \mu_j = \frac{1}{n}\sum_{i=1}^{n}\mathbf{H}_{ij}, \tag{12}$$

where $\epsilon$ is a small value to prevent division by zero. The initial representations $\mathbf{H}$ are distributed on an arbitrary space characterized by a mean center $\mu = \{\mu_1, \ldots, \mu_K\}$. After batch normalization, the space of the transformed representations $\mathbf{H}'$ is standardized to a space with zero mean and unit variance. This transformation effectively constitutes a center-away displacement, i.e., each feature vector is shifted away from the center of the original distribution. Moreover, Batch Normalization ensures that each feature dimension exhibits a variance of unity, thereby facilitating a more uniform distribution across the transformed representations. Although the primary objective of batch normalization is not representation scattering, it can be inferred from the aforementioned analysis that BN's adjustment of data distribution indirectly facilitates a more uniform scattering of representations across space. $\qquad\square$

### D.3 Proof of Theorem 4

*Proof.* It is well-known that the InfoNCE loss (with $\ell 2$ norm) is enforcing the embeddings to be uniformly distributed in a hypersphere. It is exactly representation scattering. We begin with analysing the formula of the InfoNCE [14] loss:

$$\mathcal{L}_{\text{InfoNCE}}(\mathbf{h}_i) = -\log\left(\frac{e^{\theta(\mathbf{h}_i, \mathbf{h}_i')}}{e^{\theta(\mathbf{h}_i, \mathbf{h}_i')} + \sum_{k=1, k\neq i}^{n}(e^{\theta(\mathbf{h}_i, \mathbf{h}_k)} + e^{\theta(\mathbf{h}_i, \mathbf{h}_k')})}\right), \tag{13}$$

where $\mathbf{h}_i$ denotes the representation of node $v_i$, obtained from one augmented view of the original graph, and $\mathbf{h}_i'$ represents the representation of the corresponding node $v_i'$ from the other augmented view. Here, $\theta(\mathbf{h}_i, \mathbf{h}_i') = \text{Sim}(\mathbf{h}_i, \mathbf{h}_i')/\tau$, where $\text{Sim}(\cdot)$ denotes the cosine similarity between node representations. Specifically, $\text{Sim}(\mathbf{h}_i, \mathbf{h}_i')$ is computed as the normalized dot product: $\text{Sim}(\mathbf{h}_i, \mathbf{h}_i') = (\mathbf{h}_i^\top \mathbf{h}_i')/(\|\mathbf{h}_i\|\|\mathbf{h}_i'\|)$. Eq. 13 states that minimizing the loss function requires the maximization of similarity among positive samples and the simultaneous minimization of similarity among negative samples.

To simplify the proof, we define $\theta(\mathbf{h}_i, \mathbf{h}_i')$ as the cosine similarity between node pairs, omitting the hyper-parameter $\tau$. Furthermore, we replace the logarithm with the natural logarithm $\ln$ in Equation (13), yielding the expression below:

$$\mathcal{L}_{\text{InfoNCE}}(\mathbf{h}_i) = -\theta(\mathbf{h}_i, \mathbf{h}_i') + \ln\sum_{j=1}^{n}\left(e^{\theta(\mathbf{h}_i, \mathbf{h}_j)} + e^{\theta(\mathbf{h}_i, \mathbf{h}_j')}\right). \tag{14}$$

To facilitate a clearer analysis of Equation (14) and given the convex nature of the exponential function, we simplify the expression using Jensen's Inequality:

$$\frac{1}{2n}\sum_{j=1}^{2n}e^{\theta_j} \geq e^{\frac{1}{2n}\sum_{j=1}^{2n}\theta_j}, \tag{15}$$

where $\theta_j$ represents the cosine similarity between the anchor node $v_i$ and other nodes that form negative pairs. According to Eq. (15), the sum of exponentials can be transformed into the exponential of a sum, which can then cancel out with the logarithm, thereby simplifying the equation:

$$\mathcal{L}_{\text{InfoNCE}}(h_i) \geq \frac{1}{2n}\sum_{j=1}^{2n-1}\theta_j + \ln(2n). \tag{16}$$

In Eq. (16), it is demonstrated that the InfoNCE loss associated with the anchor node $v_i$ serves as an upper bound to the sum of cosine similarities between $v_i$ and all negative samples. To elucidate the relationship between representation scattering and InfoNCE loss more clearly, the following lemma is presented:

**Lemma 6.** *Let $v_i$ be the anchor node, and let $\{v_1, v_2, v_3, \cdots, v_n\}$ be the set of nodes, excluding $v_i$. The average of the sum of cosine similarities between $v_i$ and all other nodes in the set is equivalent to the cosine similarity between the $\ell_2$-normalized vector of $v_i$ and the $\ell_2$-normalized mean vector obtained from all other nodes.*

The proof is provided in Appendix D.4. Lemma 6 demonstrates that the minimization of Eq. (13) for the anchor node $v_i$ entails a concurrent reduction in the cosine similarity between the $\ell_2$-normalized vector of $v_i$ and the $\ell_2$-normalized centroid of all other nodes. This process can be viewed as transforming the set of nodes onto a hypersphere with a radius of 1 and distancing the anchor node from the centroid of other nodes. We consider this hypersphere as subspace $\mathbb{S}^k$, and the centroid as the scattered center $\mathbf{c}$, which aligns with our definition of representation scattering. Thus, minimizing the InfoNCE loss effectively promotes representation scattering. $\square$

### D.4 Proof of Lemma 6

*Proof.* Given two vectors $\mathbf{v_i}$ and $\mathbf{v_j}$, their cosine similarity is defined as $\frac{\mathbf{v}_i \mathbf{v}_j}{\|\mathbf{v}_i\|\|\mathbf{v}_j\|}$, which is the cosine of the angle between them

$$\sum_{j=1,j\neq i}^{2n-1} \theta_{ij} = \sum_{j=1,j\neq i}^{2n-1} \frac{\mathbf{v_i}\mathbf{v_j}}{\|\mathbf{v}_i\|\|\mathbf{v}_j\|}. \tag{17}$$

For $\ell_2$-normalized vectors $\hat{\mathbf{v_i}} = \frac{\mathbf{v_i}}{\|\mathbf{v}_i\|}$, this is equivalent to their dot product: $\hat{\mathbf{v_i}} \cdot \hat{\mathbf{v_j}}$:

$$\sum_{j=1,j\neq i}^{2n-1} \frac{\mathbf{v_i}\mathbf{v_j}}{\|\mathbf{v}_i\|\|\mathbf{v}_j\|} = \sum_{j=1,j\neq i}^{2n-1} \hat{\mathbf{v_i}} \cdot \hat{\mathbf{v_j}}. \tag{18}$$

Since the dot product of vectors possesses the distributive property of multiplication:

$$\sum_{j=1,j\neq i}^{2n-1} \hat{\mathbf{v_i}} \cdot \hat{\mathbf{v_j}} = \hat{\mathbf{v_i}} \cdot \sum_{j=1,j\neq i}^{2n-1} \hat{\mathbf{v_j}}. \tag{19}$$

Finally, we compute the mean of Eq. 19 :

$$\frac{1}{2n}\sum_{j=1,j\neq i}^{2n-1} \theta_{ij} = \hat{\mathbf{v_i}} \cdot \frac{1}{2n}\sum_{j=1,j\neq i}^{2n-1} \hat{\mathbf{v_j}} = \hat{\mathbf{v_i}} \cdot \mathbb{E}_{v_j \in \mathcal{V}, j\neq i}(\hat{\mathbf{v_j}}). \tag{20}$$

As a consequence, the mean of the sum of cosine similarity between the anchor node $v_i$ and all other nodes can be regarded as the expected similarity of the $\ell_2$-normalized $v_i$ with all other $\ell_2$ normalized points. $\square$

## E  Related Work

Graph Representation Learning (GRL) has attracted substantial attention in recent years due to the broader application of graph data across various real-world scenarios [35]. The primary goal of GRL is to embed graph data, which is inherently high-dimensional, sparse, and non-Euclidean, into a lower-dimensional, dense, Euclidean space to facilitate easier processing in downstream tasks. Early GRL works rely on Matrix Factorization [36] and Random Walk [37], failing to simultaneously utilize the structural and attribute information of graphs [38]. In recent years, Graph Neural Networks (GNNs) have achieved great success, such as GCN [18], GAT [19], GraphSAGE [39], which effectively extract graph attribute and structural information simultaneously through message passing and aggregation mechanisms. However, the training of GNNs requires a large amount of manually annotated labels, which is expensive and labor-intensive [6]. To overcome this limitation, Graph Contrastive Learnings (GCLs) have emerged as a promising and increasingly popular learning paradigm for handling unlabeled graph data.

Currently, there are three mainstream graph contrastive frameworks, each inspired by methods from visual contrastive learning. DGI [7], inspired by Deep InfoMax (DIM) [40], generates a noise distribution through random shuffling and learns node representations by maximizing the Jensen-Shannon (JS) divergence between the semantic distribution of the original graph and the noise distribution. GRACE [8], a representative method based on InfoNCE loss and inspired by SimCLR [41], generates two correlated graph views through augmentation techniques. This framework employs the InfoNCE loss to enhance the similarity of identical nodes across the two views while minimizing the similarity among different nodes. BGRL [10], inspired by BYOL [42], utilizes two

distinct encoders: an online encoder and a target encoder. The online encoder is trained to predict the output of the target encoder, while the target encoder is updated by Exponential Moving Average (EMA).

Although the three mainstream graph contrastive learning frameworks: GRACE, DGI, and BGRL, have achieved notable success in addressing the challenges of unlabeled graph data, the underlying mechanisms driving their success have not been fully explored. Current research predominantly focuses on the implementation and optimization of these frameworks [6, 9, 12, 13], with insufficient understanding of their theoretical foundations and intrinsic principles. In visual contrastive learning, studies have revealed that "uniformity", which refers to the uniform distribution of representations in hypersphere, is a key factor contributing to improved generalization and discriminative power of models [17]. However, due to the non-Euclidean structure of graphs and the complex dependencies among nodes, this theory cannot be directly applied to graph contrastive learning. The unique characteristics of graphs necessitate a reevaluation and adaptation of the uniformity concept to develop a corresponding theoretical framework suitable for graphs. In this paper, we reanalyze the three mainstream graph contrastive learning frameworks and propose a new mechanism: representation scattering, which is prevalent and important in these frameworks. Furthermore, considering the interconnected nature of graphs, we design a constraint mechanism to achieve an adaptive representation scattering.

## F   A Detailed Discussion on the Limitations

- **Application of Heterogeneous Datasets.** In this work, we focused exclusively on homogeneous datasets and achieved satisfactory results. Our experiments demonstrated the effectiveness of the proposed topology-based constraint under the assumption of homogeneity. However, this constraint may face challenges when applied to heterogeneous datasets. Heterogeneous datasets often exhibit varying structures and characteristics, which could undermine the performance of our method. Consequently, addressing these challenges and adapting our approach to handle heterogeneous datasets effectively is a crucial direction for future research. We plan to investigate and refine our method to ensure its robustness and applicability across diverse and complex datasets.

- **The Methods of Balancing RSM and TCM.** RSM aims for the representations of nodes to be scattered from each other, while TCM prevents excessive scattering among nodes with the same semantics. These two mechanisms inherently have an adversarial relationship. We use EMA to balance this adversarial relationship, which is just one of many possible methods.

## G   Broader Impacts of SGRL

As a preliminary exploration in the field of GCL, SGRL has the following impacts:

- Uncovering potential success factors in GCL: SGRL explores three different mainstream graph contrastive methods and identifies their common underlying factors, providing a new perspective for future GCL research.

- Applications in many fields: Compared to existing methods, SGRL offers higher accuracy and can more efficiently process graph data. It can be widely applied in various fields such as social networks, biological networks, and recommendation systems.

- Negative social impacts: There are no negative social impacts foreseen.

