# OpenReview forum: "Exploitation of a Latent Mechanism in Graph Contrastive Learning: Representation Scattering"
_NeurIPS.cc/2024/Conference — NeurIPS 2024 oral_

### Official Review · Reviewer_6M1t · 2024-07-11

**Soundness:** 4
**Presentation:** 3
**Contribution:** 4
**Rating:** 9
**Confidence:** 4

**Summary:**

The authors focus on exploring an essential mechanism existing in different contrastive strategies. They first define a concept in embedding space, which contains a center $\mathbf{c}$, a subspace $\mathbb{S}$ and two constraints, called representation scattering. They then investigate the relationships between this concept and the mainstream GCL frameworks, through intuitive experiments and rigorous formula derivation. These discoveries motivate the authors to develop a new GCL framework aligned with this concept. The authors also present a model, namely SGRL, which includes a contrastive loss that directly makes representations away from the mean center. Experiments validate that SGRL has a better performance.

**Strengths:**

* As the paper states, existing work typically lacks a deeper insight into the mechanism behind various graph contrastive frameworks, which is a very important issue in GCL. This paper makes a positive contribution towards achieving this goal.
* The proposed method SGRL is technically sound. The authors introduce the representation scattering mechanism and design SGRL following this new mechanism. Each section is supported by some convincing theorems and derivations.
* The writing of this paper is clear and this work is comprehensive. The authors provide extensive experiments and appendices to substantiate their claims.

**Weaknesses:**

* The authors demonstrate that the representation scattering mechanism exists in several graph contrastive frameworks and argue that these methods do not fully utilize this mechanism. But the reviewer only observes discussions about the shortcomings of several baselines in the paper, lacking a deeper discussion on how these methods do not fully utilize representation scattering.
*  In the viewpoints of the reviewer, the constraint based on topological aggregation seems unnecessary. Since the encoder already aggregates information from neighbors, adding an additional loss may lack a sufficient justification.
* The SGRL employs two encoders with non-shared parameters, while the authors provide little explanation on this point. Based on the experience of the reviewer, allowing the two encoders to share parameters might perform better, as it may prevent divergences in the learned patterns between the models.

**Questions:**

In this work, the theories and methods proposed by the authors are mainly based on node-level tasks. So, the reviewer wonders whether this approach can achieve the same powerful performance in graph-level tasks.

**Limitations:**

The authors have discussed the limitations of their work adequately. I have no further concerns.

---

> ### Author Rebuttal · Authors · 2024-08-06
>
> We thank the reviewer for the valuable suggestions and comments. We respond below.
>
> > The authors demonstrate that the representation scattering mechanism exists in several graph contrastive frameworks and argue that these methods do not fully utilize this mechanism. But the reviewer only observes discussions about the shortcomings of several baselines in the paper, lacking a deeper discussion on how these methods do not fully utilize representation scattering.
>
> Thank you for your comment. We have addressed your concern in Section 3. Here, we provide further explanation. First, most existing GCLs are mainly improved based on one of the following framework: the InfoNCE-based framework, the DGI framework, and the BGRL framework. As described in Definition 1, Representation Scattering is a simple and effective mechanism. However, the three frameworks and the methods based on them have not recognized for its importance, resulting in limited performance and low efficiency.
>
> **InfoNCE-based methods:** InfoNCE loss function indirectly achieves representation scattering by separating node pairs apart. It treats all negative samples indiscriminately and doesn’t differentiate their respective contributions to the loss. Moreover, it evaluates similarities for every possible pair within the batch, giving rise to $O(n^2)$ complexity.
>
> **DGI-like methods:** The objective of DGI-like methods is to maximize the Jensen-Shannon divergence between the original graph and the perturbed graph, which is a special case of representation scattering. However, the perturbed graph is unnecessary, leading to extra memory overhead and potential bias in negative samples.
>
> **BGRL-like methods:** The key component of BGRL-like methods, Batch Normalization (BN), is a special case of representation scattering. However, BN is not used in training; it is merely used to adjust the distribution in the embedding space. This could result in sub-optimal performance.
>
> > In the viewpoints of the reviewer, the constraint based on topological aggregation seems unnecessary. Since the encoder already aggregates information from neighbors, adding an additional loss may lack a sufficient justification.
>
> Thanks for the comment, but we disagree respectfully. Topology-based Constraint Mechanism (TCM) is very important because it enables representations to be scattered more precisely in the embedding space, significantly improving performance.
> 1) GNNs primarily aggregate local information and fail to effectively capture common features among local nodes (especially with fewer layers). In contrast, TCM enables node representations to be scattered globally while being aggregated locally in the embedding space.
>
> 2) The introduction of TCM significantly enhances the capability of SGRL to process graphs. Different from image and text data, the linked nature of nodes results in connected nodes often being closer in the embedding space. The lack of TCM could lead to semantically similar nodes being distant and semantically dissimilar nodes being close.
>
> 3) The ablation study of TCM is presented in Table 3 of our paper. The accuracy of SGRL-TCM decreases by 0.5% on average, with a notable reduction of approximately 1% on Wiki-CS. This further validates the significance of TCM.
>
> We will add this discussion in future version.
>
> > The SGRL employs two encoders with non-shared parameters, while the authors provide little explanation on this point. Based on the experience of the reviewer, allowing the two encoders to share parameters might perform better, as it may prevent divergences in the learned patterns between the models.
>
> Admittedly, in many GCL frameworks, such as GRACE and DGI, shared-parameter encoders are employed to maintain consistency in node representations across different views. However, this approach does not apply to SGRL. To achieve adaptive representation scattering, we have designed two distinct mechanisms, RSM and TCM. It is important to note that the objectives of these two mechanisms can be somewhat conflicting. Shared-parameter encoders might lead to decreased performance and efficiency due to the following reasons:
>
> **Conflicting Objectives:** The encoders must satisfy two potentially conflicting objectives simultaneously during training, which can lead to unstable outcomes.
>
> **Increased Iterations:** More iterations may be necessary to find a set of parameters that effectively balances both objectives.
>
> To address these challenges, we utilize two separate encoders for the distinct mechanisms and balance the potentially conflicting objectives through Exponential Moving Average (EMA). This approach is more efficient and results in smoother performance.
>
> > In this work, the theories and methods proposed by the authors are mainly based on node-level tasks. So, the reviewer wonders whether this approach can achieve the same powerful performance in graph-level tasks.
>
> Thanks for this thought-provoking question. In this paper, we revisit GCL frameworks that focus on node-level tasks. There are some similarities between graph-level and node-level tasks in terms of achieving representation scattering:
>
> **Scattered Center Definition:** In node-level tasks, the scattered center is defined by computing the mean of all node representations. Similarly, for graph-level tasks, the mean of all sub-graph representations can be used to define the graph-level scattered center. This approach allows us to achieve representation scattering through a center-away loss.
>
> **Application of TCM:**  Although we can't constrain the graph representations as we do with nodes, the similarity between different graphs can be evaluated by designing appropriate graph kernel functions. Therefore, we can still design a graph-level "TCM" to constrain the representations.
>
> We plan to further explore the application of representation scattering in graph-level tasks in future work.

---

> ### Comment · Reviewer_6M1t · 2024-08-14
>
> The authors have provided thorough and insightful responses to my previous questions, which have greatly improved the clarity of the manuscript. As I noted in my initial evaluation, the work presents a novel idea, demonstrates high readability, and is presented in a clear and accessible manner. Additionally, the experimental studies are robust and well-executed. I have no further concerns and am therefore inclined to raise my score and strongly recommend this work.

---

### Official Review · Reviewer_dtE3 · 2024-07-11

**Soundness:** 4
**Presentation:** 4
**Contribution:** 3
**Rating:** 9
**Confidence:** 5

**Summary:**

In this paper, the authors provide an interesting discovery: the successes with mainstream GCL paradigms essentially come from implicitly scattering representations. They point out that the bottleneck of current GCLs lies in ignoring this, and they provide detailed theoretical proofs. Furthermore, they propose a new method to fully utilize representation scattering. They propose an asymmetric framework that, through carefully designed central discrete loss, enhances the distinctiveness of representations, resulting in favorable performance improvements.

**Strengths:**

1. The paper is well-written and easy to follow.
2. The authors observe an interesting phenomenon: intuitively, DGI-based methods and InfoNCE-based methods seem conflicting on node-level proxy tasks, yet both achieve good performance. Previous work has overlooked the connection between them, but this paper unifies these paradigms, potentially providing insights for the future development of graph representation learning.
3. Exploring and formalizing the definition of representation scattering is highly valuable. Previously, the notion of uniformity—intuitively understood as diversity in encoding negative samples—was only mentioned in graph studies based on InfoNCE. However, this paper provides a clear definition and broadens its application across all GCL paradigms.

**Weaknesses:**

1. As I mentioned in the summary, the authors have designed an asymmetric contrastive framework with two opposing types of losses, which may lead to misunderstandings of the training process. Although the description in the paper is given, including an algorithm flowchart would be better.
2. In Figure 1, the authors plot the t-SNE visualization of DGI on Co.CS . When the number of encoder layers changes, two distinctly different results are produced. I hope the authors provide a detailed explanation of this phenomenon.
3. Is the proposed central discrete loss effective in all scenarios? In extreme cases, if all node representations converge at a single point, can this loss achieve dispersion?

**Questions:**

see weaknesses

**Limitations:**

see above

---

> ### Author Rebuttal · Authors · 2024-08-07
>
> Thank you for taking the time to read and review our submission. We have provided our responses below.
>
> > As I mentioned in the summary, the authors have designed an asymmetric contrastive framework with two opposing types of losses, which may lead to misunderstandings of the training process. Although the description in the paper is given, including an algorithm flowchart would be better.
>
> Thank you for your valuable suggestions. We apologize for any misunderstanding. To facilitate reader understanding, we will provide an algorithm flowchart in future version. Here, please allow us to clarify the structure of the SGRL algorithm.
>
> Given a graph $\mathcal{G}$, two different encoders $f_\theta(\cdot)$ and $f_\phi(\cdot)$ generate node embeddings $H_{online}$  and $H_{target}$ , respectively.
>
> - **Topological Aggregation:** $H_{online}$ is processed through TCM to obtain topologically aggregated representations $H_{online}^{topology}$, without updating the parameters of $f_\theta(\cdot)$ and $f_\phi(\cdot)$.
>
> - **Representation Scattering:** For $H_{target}$, we use RSM to encourage node representations to diverge from the center $c$, and update the parameters of $f_\phi(\cdot)$.
>
> - **Prediction and Parameter Update:** $H_{online}^{topology}$ is used to predict $H_{target}$ using the predictor $q_{\theta}$. During this process, the parameters of $f_\theta(\cdot)$ are updated while the gradient of $f_\phi(\cdot)$ is stopped. Finally, we gradually update the parameters of $f_\phi(\cdot)$ using Exponential Moving Average (EMA).
>
> To facilitate reader understanding, we will provide an algorithm flowchart in future version.
>
> > In Figure 1, the authors plot the t-SNE visualization of DGI on Co.CS . When the number of encoder layers changes, two distinctly different results are produced. I hope the authors provide a detailed explanation of this phenomenon.
>
> We have provided explanations in the caption of Figure 1, but your comments made us realize that further clarification is needed.
>
> **Figure 1(a):** This figure shows the t-SNE embedding of randomly initialized GNN on Co.CS, where blue points represent the representation distribution of the perturbed graph and red points depict that of the original graph. It can be clearly observed that the representation distribution of the perturbed graph approximates the center of the original graph's representation distribution, which is consistent with the formulation provided in Theorem 1.
>
> **Figure 1(b):** This figure illustrates the t-SNE embedding after training on a one-layer GNN. Compared to Figure 1(a), there is a more pronounced separation between the specific semantic distributions of the original graph (red points) and the center, while the distribution of the perturbed graph (blue points) has converged more towards the center. Additionally, the intra-class boundaries within the original graph have become more distinct. The transition from Figure 1(a) to Figure 1(b) intuitively shows the outcome of DGI training, where the specific semantic distribution of the original graph diverges from the center. According to Corollary 3, this process is a specific case of representation scattering.
>
> **Figure 1(c):** This figure presents the result after training on a two-layer GNN. The incorporation of nonlinear activation functions makes Figure 1(c) align closely with the objective of DGI, which is to maximize the Jensen-Shannon (JS) divergence between the distributions of the original and perturbed graphs. Based on Figures 1(b) and 1(c), we can conclude that the training objective of DGI maximizes the JS divergence between the specific semantic distribution of the original graph and its mean distribution.
>
> > Is the proposed central discrete loss effective in all scenarios? In extreme cases, if all node representations converge at a single point, can this loss achieve dispersion?
>
> The extreme scenario you mentioned, where all node representations converge to a single point, is indeed meaningless as it would result in indistinguishable node representations. We have considered a scenario similar yet meaningful to the one you described.
> Consider an embedding subspace $\mathbb{E}$ containing a set of points $\mathcal{V}$ in $\mathbb{R}^n$. In this subspace $\mathbb{E}$, for any $v_i, v_j \in \mathcal{V}$, their representations satisfy $||h_i - h_j||_2^2 < \epsilon$. In this case, the center-away loss plays a significant role. It emphasizes distancing nodes from a relative mean center rather than a fixed absolute center. Initially, even if all nodes are clustered within the subspace $\mathbb{E}$, by promoting their distance from the dynamic scattered center $\mathbf{c}$, SGRL is still effective.

---

### Official Review · Reviewer_ErfB · 2024-07-11

**Soundness:** 4
**Presentation:** 3
**Contribution:** 4
**Rating:** 8
**Confidence:** 4

**Summary:**

This paper provides an insightful perspective of representation scattering to unify various GCL frameworks, and proposes an effective framework called SGRL. Specifically, the contributions are as follows: 1) Theoretically, with a well-defined representation scattering concept, the authors provide a universal theoretical explanation for the success of existing GCL frameworks. 2) They propose SGRL, which employs a unique adversarial approach to effectively utilize this mechanism. Specifically, SGRL integrates topological aggregation with representation scattering, with two adversarial objectives smoothed by EMA. 3) The proposed SGRL achieves powerful performances in extensive experiments across multiple tasks.

**Strengths:**

1. The theoretical foundation is solid. The authors provide a new concept of representation scattering with a clear mathematical definition. All theoretical claims have formal and rigorous proofs. In addition, appropriate motivation experiments are provided to support their theorems.
2. The authors design a well-motivated and novel framework. Through a comprehensive exploration of representation scattering, SGRL seems break through the limitations of existing methods and can be regarded as a new branch of GCL. Besides, SGRL integrates topological aggregation with representation scattering via adversarial strategy, which is interesting and technically reasonable.
3. SGRL shows strong performance across various tasks on multiple datasets.

**Weaknesses:**

1. Although the results in Figure 5 clearly demonstrate how model performance varies with different strengths of topological constraints, I believe it would be beneficial for the authors to present more results from additional datasets. To my knowledge, the preprocessing method used for Wiki-CS often differs from that of the other four datasets, which may influence the choice of $k$. Therefore, I would appreciate it if the authors could provide further experiments to enrich their analysis.
2. In Definition 1, there are two constraints in representation scattering, the center-away constraint and the uniformity constraint. When considering only the goal of making representations scattered enough, the center-away constraint seems somewhat redundant, as satisfying the uniformity constraint alone can ensure the discreteness of representations. The authors should provide additional explanations regarding the role of the center-away constraint in the process of representation scattering.
3. I suggest that the authors provide more detailed explanations of some symbols, even though this is common in graph representation learning. For example, the $\alpha_{ij}$ and $d_i$ in Equation 1.

**Questions:**

See the above weakness.

**Limitations:**

Yes, they have.

---

> ### Author Rebuttal · Authors · 2024-08-06
>
> Thanks for your time in reading and reviewing our submission. We respond below.
>
> > Although the results in Figure 5 clearly demonstrate how model performance varies with different strengths of topological constraints, I believe it would be beneficial for the authors to present more results from additional datasets. To my knowledge, the preprocessing method used for Wiki-CS often differs from that of the other four datasets, which may influence the choice of
> . Therefore, I would appreciate it if the authors could provide further experiments to enrich their analysis.
>
> Thank you for the valuable suggestion. In Section 5.2, we conduct a sensitivity analysis of $k$ and show the results in Figure 5, showing the impact of topological constraint. Your comments have made us realize that we need to include more results from datasets like Wiki-CS, which utilize a unique pre-processing method, to help better understand this mechanism. To enhance the readability of our paper, we present additional experimental results from other datasets in Table Re-ErfB-1. Admittedly, different pre-processing methods may influence the peak of SGRL, but they do not affect the analysis presented in our paper.
>
> **Table Re-ErfB-1: Additional Hyper-parameter Analysis on $k$.**
> || 0 | 1 | 2 | 3 | 4 | 5 | 6 | 7 | 8 | 9 | 10 |
> |-|-|-|-|-|-|-|-|-|-|-|-|
> | Wiki-CS | 78.56±0.05 | 79.40±0.10 | 79.48±0.01 | **79.54±0.03** | 79.53±0.01 | 79.45±0.04 | 79.37±0.03 | 79.28±0.06 | 79.21±0.06 | 79.20±0.04 | 79.11±0.05 |
> | Amazon.Photo | 93.54±0.05 | **93.95±0.03** | 93.84±0.02 | 93.83±0.02 | 93.80±0.01 | 93.76±0.02 | 93.73±0.01| 93.73±0.02 | 93.69±0.05 | 93.72±0.05 | 93.71±0.05 |
> | Co.Physics | 96.16±0.03 | **96.23±0.01** | 96.18±0.02 | 96.18±0.03 | 96.18±0.03 | 96.17±0.02 | 96.17±0.01 | 96.17±0.01| 96.16±0.02 | 96.18±0.01 | 96.18±0.01 |
>
> The results are still consistent with the analysis in the paper: considering the differences in the degree of scattering of different node representations is necessary and the hyper-parameter $k$ exhibits an unimodal effect on SGRL's performance. As $k$ increases, the model performance initially improves, indicating that topological constraint is effective. With the continuous increase of $k$, exceeding its peak will lead to a decline in performance. This is due to excessive constraints intensifying the antagonism between TCM and RSM, leading to RSM failure. Overall, although the pre-processing methods of different datasets may be different, this does not affect our conclusion. We will include this discussion in future version.
>
> > In Definition 1, there are two constraints in representation scattering, the center-away constraint and the uniformity constraint. When considering only the goal of making representations scattered enough, the center-away constraint seems somewhat redundant, as satisfying the uniformity constraint alone can ensure the discreteness of representations. The authors should provide additional explanations regarding the role of the center-away constraint in the process of representation scattering.
>
> In fact, the center-away constraint is a vital component of the representation scattering mechanism. The reasons are as follows.
> - **Clarification of Scattering Mechanism:** With this constraint, Definition 1 delineates more precisely the scattering mechanism inherent in three frameworks (InfoNCE, DGI, BGRL), dictating how node representations achieve scattering within the embedding space. Despite their diverse approaches, all these frameworks incorporate a principle of moving away from the center.
>
> - **Expressiveness of Representations:** If node representations are close to the center, the expressiveness of representations will be weakened. For instance, in BGRL-like methods, the lack of center-away constraint will result in a concentration of the representations near the center, which diminishes the informativeness of the embeddings and reduces their distinctiveness. This is also a drawback of Batch Normalization (BN).
>
> Therefore, we introduce center-away constraint to ensure the completeness of the representation scattering theory. We will provide more explanations on this aspect in the appendix to complete the theorem.
>
> > I suggest that the authors provide more detailed explanations of some symbols, even though this is common in graph representation learning. For example, the $\alpha_{ij}$ and $d_i$ in Equation 1.
>
> In Equation 1, $\alpha_{ij}$ represents the normalized connection weight between node $i$ and its neighboring node $j$, typically calculated as the element $A_{ij}$ from the adjacency matrix $A$ divided by the degree $d_i$ of node $i$, i.e., $\alpha_{ij} = \frac{A_{ij}}{d_i}$. We will supplement this explanation in future version.

---

> > ### Comment · Reviewer_ErfB · 2024-08-14
> >
> > Thanks for the response. After reading the authors’ response as well as the other reviewers’ comments, my main concerns, particularly regarding the center-away constraint, have been addressed. I appreciate the interesting idea of this paper, and thus would like to increase my rating to 8.

---

### Official Review · Reviewer_HnqZ · 2024-07-14

**Soundness:** 4
**Presentation:** 4
**Contribution:** 4
**Rating:** 9
**Confidence:** 5

**Summary:**

The authors attempt to propose a universal theory of graph contrastive learning which may benefit this field. Most existing GCLs directly inherit from other fields. While existing GCLs have achieved similar success, there are intuitive differences and even conflicts in the operations. By analyzing three representative GCL frameworks from the unique perspectives of topology and message passing, the authors of this work find a key factor behind GCLs, which is defined as representation scattering. To better achieve node representations scattering, they mine the natural linking characteristic of graphs and propose a new contrastive concept. It involves contrasting nodes with embedding centers and implementing aggregation constraints based on graph topology, which replaces the traditional inefficient augmentation and sampling-based GCL proxy tasks. The proposed method is compared to existing GCL methods across multiple downstream scenarios to demonstrate the superiority, confirming it can learn high-quality node representations in a self-supervised manner.

**Strengths:**

1. The authors propose a new theory that unifies existing GCLs, which is insightful and may have implications for a broad field.

2. The presentation of the paper is clear and easy to understand.

3. The theoretical analysis is comprehensive and rigorous. This reveals the underlying mechanism of the success of existing GCLs, and gives strong theoretical support to the proposed method.

4. The empirical evaluation is sufficient, showing powerful performances across various datasets.

Overall, I think this paper makes a significant contribution to the field of graph representation learning.

**Weaknesses:**

1. The proposed SGRL is an augmentation-free framework, avoiding manual bias and reducing training overhead in augmentation. To my knowledge, there are also some augmentation-free contrastive methods available, such as [1, 2]. While data augmentation is not the main focus of this paper, it may be beneficial if the authors provide a comparison between SGRL and this type of method.

2. In SGRL, an existing component, EMA [3, 4], is given new functionality to balance the adversarial effects of two constraints (RSM and TCM) and achieve satisfactory results. I find this innovation interesting. However, it would be better if the authors provided further discussions and experiments on the chosen EMA’s hyper-parameters, since different hyper-parameters of EMA may make the two adversarial branches achieve different equilibrium.

3. The provided theory primarily discusses the impact of representation scattering mechanisms on various GCLs, encompassing multiple aspects such as data augmentation, GNN encoders, and negative contrasting. However, the authors seem to have omitted positive contrasts. While topological aggregation constraints can be seen as a more effective form of positive sample contrast, I still hope the authors can incorporate positive contrast into the proposed theory.

Some minor points:

1. p. 2 line 72 - I am not sure what is meant by "learning invariance post-disturbance."

2. p. 3 line 149 - Figure 1 is mentioned too frequently within a single section.

3. Table 1: Some names of the datasets used have prefixes (Co.CS and Co.Physics) while others do not (Computers and Photo). It would be better to maintain a consistent format.

[1] Lee N, Lee J, et al., Augmentation-free self-supervised learning on graphs, AAAI, 2022.

[2] Yu J, Yin H, et al., Are graph augmentations necessary? simple graph contrastive learning for recommendation, SIGIR, 2022.

[3] Thakoor S, Tallec C, et al., Large-scale representation learning on graphs via bootstrapping, ICLR, 2022.

[4] Grill J B, Strub F, et al., Bootstrap your own latent-a new approach to self-supervised learning, NeurIPS, 2020.

**Questions:**

The analysis of DGI is based on the assumption that the perturbation is randomly shuffling the node features. While the authors have provided discussions in Theorem 1 when this assumption fails, I wonder whether the discussion on the shortcomings of DGI-like methods remains valid when this assumption fails.

---

> ### Author Rebuttal · Authors · 2024-08-07
>
> We thank the reviewer for the time in reading our paper and giving valuable suggestions. To address your concerns, we respond below.
>
> > The proposed SGRL is an augmentation-free framework, avoiding manual bias and reducing training overhead in augmentation. To my knowledge, there are also some augmentation-free contrastive methods available, such as [1, 2]. While data augmentation is not the main focus of this paper, it may be beneficial if the authors provide a comparison between SGRL and this type of method.
>
> Thanks for your suggestion. We will provide a comparison between SGRL and [1, 2] regarding data augmentation.
>
> - [1] generates positive samples of original nodes through k-NN search and filters these samples from both local (whether nodes are connected) and global perspectives (whether they belong to the same cluster), which overemphasizes structure information.
> - [2] generates different views by introducing random, uniform noise into the original representations, neglecting the impact of the structure information.
> - In contrast, SGRL considers both structure and attribute information. Specifically, SGRL employs an asymmetric dual-channel design, generating views from a structure perspective with topological semantics ($H_{online}^{topology}$) and an attribute perspective with scattered representations ($H_{target}$). Therefore, our augmentation-free method supports a more comprehensive utilization of the graph's information.
>
> > In SGRL, an existing component, EMA [3, 4], is given new functionality to balance the adversarial effects of two constraints (RSM and TCM) and achieve satisfactory results. I find this innovation interesting. However, it would be better if the authors provided further discussions and experiments on the chosen EMA’s hyper-parameters, since different hyper-parameters of EMA may make the two adversarial branches achieve different equilibrium.
>
> Thank you for suggesting improving the experiment. To investigate the impact of EMA on the two branches, we evaluated the performance of SGRL by changing the hyper-parameter $\tau$. We set the value of $\tau$ to 0.99 (align with the value in our paper), 0.95, 0.90, 0.80, 0.50, 0.00 (SGRL-EMA), and the experimental results are shown in Table Re-HnqZ-1.
>
> **Table Re-HnqZ-1: The Impact of EMA.**
> || 0.99 | 0.95 | 0.90 | 0.80 | 0.50 | 0.00 |
> |-|-|-|-|-|-|-|
> | Co.CS | 94.15±0.04 | 94.08±0.02 | 94.11±0.02 | 94.09±0.05 | 94.04±0.01 |	93.89±0.07 |
> | Co.Physics | 96.23±0.01 | 96.18±0.05 | 96.16±0.03 | 96.16±0.01 | 96.17±0.02 |96.16±0.07 |
> | Wiki-CS | 79.40±0.13 | 79.40±0.08 | 79.38±0.08 | 79.39±0.09 | 79.38±0.06 | 79.36±0.08 |
>
> It is evident from the result that SGRL achieves the best performance when $\tau = 0.99$. This is because the topological semantic information is gradually fed into the representation scattering mechanism at each epoch, which moderates the adversarial interaction between the two branches. In addition, when the topological semantic information excessively influences the representation scattering process (as $\tau$ decreases), SGRL’s performance slightly declines but remains better than SGRL-EMA ($\tau = 0.00$). This result indicates that balancing the two adversarial branches is necessary, which is consistent with our conclusions in Section 5.2. We will explore more effective balancing methods in future work.
>
> > The provided theory primarily discusses the impact of representation scattering mechanisms on various GCLs, encompassing multiple aspects such as data augmentation, GNN encoders, and negative contrasting. However, the authors seem to have omitted positive contrasts. While topological aggregation constraints can be seen as a more effective form of positive sample contrast, I still hope the authors can incorporate positive contrast into the proposed theory.
>
> SGRL is different from traditional contrastive learning based on positive and negative sampling. In SGRL, we do not focus on defining positive and negative samples. Instead, we propose two mechanisms that train the encoder in an adversarial-like manner. In addition, positive contrasting is usually used to train the encoder to learn consistent representations across different views, whereas TCM aims to regulate the scattering of representations in space.
>
> > p.2 line 72.
>
> It indicates that TCM can enhance the robustness of SGRL. We explain this in Section 4.3.
>
> > p.3 line 149 and Table 1.
>
> Thanks for your suggestion. We will make corrections in future version.
>
> > The analysis of DGI is based on the assumption that the perturbation is randomly shuffling the node features. While the authors have provided discussions in Theorem 1 when this assumption fails, I wonder whether the discussion on the shortcomings of DGI-like methods remains valid when this assumption fails.
>
> Sorry for the confusion. We follow the setup described in Section 3.1: Node $v_i$ and its first-order neighbors follow the distribution $ p_i(\cdot)$ over $\mathbb{R}^{M+1}$. In DGI, they design the other corruption function $C$ by sampling, i.i.d., a switch parameter $\Sigma_{ij}$ to determine whether to corrupt the adjacency matrix at position $(i,j)$. Given a corruption rate $\rho$, they get perturbed graph by $\hat{A} = A \oplus \Sigma$, where $\oplus$ is the XOR (exclusive OR) operation and $\Sigma_{ij} \sim \text{Bernoulli}(\rho)$. In this case, node $v_i $ has a probability of $1/2k$ to connect with node $v_j$, where $v_i, v_j \sim p_i(\cdot)$. However, DGI indiscriminately maximizes the Jensen-Shannon divergence between the original graph and the perturbed graph, treating such cases as negative samples, which leads to additional bias. Although the probability changes (from $1/k$ to $1/2k$), this does not affect the analysis of the shortcomings of DGI-like methods.

---

> > ### Comment · Reviewer_HnqZ · 2024-08-13
> >
> > Thanks for the authors’ response which is satisfactory. In my opinion, the proposed representation scattering is rigorously proven to be a key factor in the success of existing GCLs, which may provide valuable insights for future advancements in graph representation learning. Therefore, I consider this work to be of exceptional quality.

---

### Decision · Program_Chairs · 2024-09-25

**Decision:**

Accept (oral)

**Comment:**

This paper discovers a common mechanism hidden behind main-stream GCL frameworks such as Info-NCE-based methods, DGI, and BGRL. Specifically, the authors prove that the mainstream baselines implicitly employ the representation scattering law (a combination of the center away and uniformity constraint).

This is an important contribution to the GCL community, and reviews are consistently positive. I thus recommend accepting this paper.